# Investigation of Mechanical Properties and Oil Resistance of Hydrogenated-Butadiene-Acrylonitrile-Rubber-Based Composites Across Various Temperatures

**DOI:** 10.3390/polym16233294

**Published:** 2024-11-26

**Authors:** Yu Han, Jingkai Nie, Zhanwei Zhu, Hang Yin, Lei Shi, Shuai Wang, Xiaosheng Liu, Qiang He

**Affiliations:** 1Institute of New Electrical Material, China Electric Power Research Institute Co., Ltd., Beijing 100192, China; hanyu1@epri.sgcc.com.cn (Y.H.); niejingkai1@epri.sgcc.com.cn (J.N.); liuxiaosheng@epri.sgcc.com.cn (X.L.); heqiang@epri.sgcc.com.cn (Q.H.); 2State Key Laboratory of Advanced Power Transmission Technology, Beijing 102209, China; 3State Grid Beijing Electric Power Corporation, Beijing 100031, China; zhuzhanwei@bj.sgcc.com.cn (Z.Z.); shilei@bj.sgcc.com.cn (L.S.); wangshuaik@bj.sgcc.com.cn (S.W.)

**Keywords:** rubber-based composites, HNBR, oil resistance, rheological behavior

## Abstract

The influence of molecular structure (acrylonitrile content) and formulation (carbon black and plasticizer dosage) on the rheological and mechanical properties of HNBR composites was systematically studied, with further discussion on ozone resistance and swelling behavior in transformer oil. The results demonstrated that the curing characteristics and rheological behavior of HNBR composites are closely linked to acrylonitrile content, carbon black, and plasticizer levels. Plasticizers significantly reduced the degree of crosslinking and the Payne effect, while fillers had the opposite impact. Fillers increased the modulus at 100% and 200%, reducing elongation at break, whereas plasticizers enhanced elongation at break while lowering the modulus. The effects of fillers and plasticizers on tensile strength were relatively minor. Both exhibited different influences on mechanical properties at various aging temperatures. Compression set testing revealed that under a 125 °C hot air environment, the compression set was less than 30%, while at −30 °C in cold air, it exceeded 60%. In a 125 °C hot transformer oil environment, the compression set ranged between 30% and 60%. Oil resistance tests indicated that HNBR composites with higher acrylonitrile content showed lower mass change rates in transformer oil, with further reduction achieved by increasing the plasticizer or filler content. Due to their excellent performance and resistance to ozone cracking, HNBR composites have significant potential for applications in high-altitude power grids and military-grade rubber sealing products.

## 1. Introduction

With societal advancement, rubber materials have gained increasing attention and wider applications. Hydrogenated butadiene acrylonitrile rubber (HNBR) is produced via catalytic hydrogenation of acrylonitrile butadiene rubber (NBR), during which the carbon–carbon double bonds in the NBR molecular chain are converted to single bonds, while the cyano group remains intact [1,2,3,4,5]. This transformation grants HNBR superior aging and oil resistance compared to NBR, making it widely used in the automotive industry, battery energy storage, petroleum exploration, and pipe-sealing applications [6,7,8,9,10,11,12].

In recent years, stricter environmental requirements in industrial applications, such as in sealing, have emphasized the need for rubber materials to maintain long-term sealing performance. This has made research on the aging performance of rubber materials particularly crucial [13,14,15,16,17].

HNBR tends to show performance degradation in high-temperature environments, such as reduced tensile strength. Zeng Zhang et al. [18] explored the use of nano-graphite of various sizes to improve the aging resistance of HNBR in high-temperature media. The results indicated that HNBR composites modified with 15 parts nano-graphite exhibited an aging coefficient 3.27 times higher than that of unmodified samples. Beatriz Vasconcelos et al. [19] chemically attached a Ni-P film layer to the surface of HNBR, which reduced the friction coefficient by 30–49% under a 5–10 N load compared to uncoated HNBR (1–5 N load). Jordão Gheller et al. [20] studied the impact of multi-walled carbon nanotubes on HNBR performance. Their findings showed that the addition of nanotubes shortened the vulcanization time and increased the modulus of modified HNBR by 940% compared to pure HNBR, while significantly reducing resistivity at low load levels.

Blending is a reliable approach for improving polymer properties. Jian-Liang Jiang et al. [21] prepared HNBR/AEM blends by mixing ethylene propylene acrylic rubber (AEM) with HNBR. Test results demonstrated that carbon-black-filled HNBR/AEM blends possess excellent mechanical properties, and their performance after thermal aging surpasses that of HNBR alone. HNBR is also crucial for elastic sealers in wellhead devices. Farzaneh Hassani et al. [22] investigated the thermal aging of HNBR, showing that below 150 °C, crosslinking dominates, leading to increased stiffness and reduced elongation at break. Above 150 °C, polymer chain breakage results in brittleness due to high-temperature effects.

Winoj Balasooriya et al. [23] studied the frictional properties of HNBR after swelling in IRM 903 solvent, observing a deterioration in mechanical properties, increased wear damage, but a reduced friction coefficient by about 25%. Pedro Quirinus de Ruijter et al. [24] predicted the service life of NBR and HNBR using time–temperature superposition and Arrhenius models through compression set evolution curves. Their findings showed that HNBR exhibits significantly better aging resistance than NBR, with its performance at 80 °C being at least three times higher.

This study employs mechanical blending to prepare different HNBR composites, examining the influence of molecular structure and formulation factors on the rheological and mechanical properties of HNBR composites. Additionally, it investigates the ozone resistance and swelling behavior of HNBR in transformer oil to assess its potential applications in high-altitude power grids and expand its usage in other fields.

## 2. Materials and Methods

### 2.1. Main Ingredients

The three fully saturated HNBRs with varying acrylonitrile (ACN) contents utilized in this work are HNBR21 (ACN of 21 wt%, ML (1 + 4) 100 °C of 74), HNBR34 (ACN of 34 wt%, ML (1 + 4) 100 °C of 70), and HNBR43 (ACN of 43 wt%, ML (1 + 4) 100 °C of 63), respectively, were all supplied by ARLANXEO (The Hague, the Netherlands). Carbon black (N550) was supplied by Cabot (Alpharetta, GA, USA) as filler, bis (2-(2-butoxyethoxy) ethyl) adipate (TP-95) was supplied by Rohm&Haas (Philadelphia, PA, USA) as plasticizer, dicumyl peroxide (DCP) was supplied by Arkema (Paris, France) as peroxide crosslinking agent, triallyl isocyanurate (TAIC) was supplied by Arkema (Paris, France) as cocrosslinking agent, zinc oxide (ZnO) and stearic acid were supplied by Rhein Chemie (Qingdao, China) as activating agents, and 4,4′-Dicumyl-diphenylamine (445) was supplied by Rhein Chemie (Qingdao, China) as antioxidant.

### 2.2. Formulation and Sample Preparation

This study focuses on the effects of HNBR matrix material types, carbon black, and plasticizer dosage on the performance of HNBR composites. The specific formulations are presented in Table 1 and Table 2.

First, the internal mixer (Haake Rheomix3000OS, Vreden, Germany) is set to a rotor speed of 60 rpm and a temperature of 80 °C. The HNBR matrix, carbon black, plasticizer, zinc oxide, stearic acid, and other components are added and mixed to form a masterbatch. Next, the two-roll mill (Farrel, GX-2003-GLT, Ansonia, CT, USA) is set with a roller speed ratio of 1:1.2 and a roller temperature of 30 °C. The masterbatch is added to the mill for further mixing, followed by the addition of the peroxide curing agent. All components are mixed uniformly according to the standard rubber mixing procedure to prepare the HNBR compound. Finally, test samples are prepared using a flat vulcanizing machine under a vulcanization pressure of 10 MPa, a temperature of 175 °C, and a vulcanization time of t90 + 2 min, where t90 is determined using a moving die rheometer (MDR2000, Alpha Technologies, Wilmington, DE, USA).

### 2.3. Testing Analysis

Curing Performance: The curing characteristics of HNBR composites were evaluated using a moving die rheometer (MDR 2000, ALPHA Technologies, Wilmington, DE, USA) following ASTM D2084-9. The test was conducted at 175 °C for 15 min, with a rotation angle of 0.5 degrees.

Rheological Behavior: In accordance with ASTM D660, the rheological properties of unvulcanized HNBR compound were analyzed using a rubber processing analyzer. Strain sweep tests were performed at 60 °C, with a frequency of 1 Hz and a strain range from 0.1% to 100%.

Tensile Performance: The tensile properties of HNBR composites were tested following ASTM D412, using a universal testing machine (Zwick/Roell Z005, Ulm, Germany). Dumbbell-shaped specimens were subjected to a loading speed of 500 mm/min during the test.

Low-Temperature Brittleness: The low-temperature brittleness of HNBR composites was measured according to GB/T 15256-2014 (ISO 812) [25], using the multi-sample method. Anhydrous ethanol served as the heat transfer medium, and type B samples were used.

Oil Aging Performance: HNBR composites were immersed in transformer oil at 125 °C for oil resistance testing. The oil aging performance was evaluated in accordance with GB/T 1690-2010 [26] and GB/T 7759.1-2015 [27].

Ozone Aging Resistance: Ozone resistance was tested based on GB/T 7762-2014 [28]. The test conditions included a static elongation of 20%, an ozone concentration of 500 pphm, a duration of 16 h, and a temperature of 40 °C.

## 3. Results and Discussion

### 3.1. Curing and Rheological Properties of HNBR Composites

First, the vulcanization characteristics of HNBR composites were examined to investigate the effects of the HNBR matrix type, fillers, and plasticizers on vulcanization behavior, as shown in Figure 1 and Figure 2.

The structural differences in the rubber macromolecular chain lead to varying changes in the vulcanization characteristics of HNBR composites. As shown in Figure 1, HNBR34, which has a lower ACN content, exhibits higher torque, while HNBR43 displays lower vulcanization torque. Additionally, incorporating HNBR21, which also has a lower ACN content, into the HNBR composite material results in a slight increase in vulcanization torque.

Figure 1A,B illustrate that increasing the filler content (from N5P5 to N6P5) raises both the maximum and minimum torque of the HNBR composites. This increase is attributed to the enhanced interaction between the rubber macromolecular chains and carbon black due to the higher filler content.

The effect of plasticizers on vulcanization torque is even more pronounced. Adding five parts of plasticizer (from P0 to P5) significantly reduces both the maximum and minimum torque. The decrease in minimum torque indicates a reduction in the viscosity of the HNBR composite, thereby enhancing its processing performance. The reduction in maximum torque occurs because the plasticizer molecules occupy spaces between the HNBR macromolecular chains, diminishing their interaction and weakening the bond between the rubber and filler.

The difference between the maximum torque (MH) and the minimum torque (ML) (MH−ML) reflects the total crosslinking degree of sulfur-cured rubber (both physical and chemical crosslinking), as shown in Figure 2. The torque difference of the HNBR34 composite (H3 series) is greater than that of the HNBR43 composite (H4 series). This indicates that, under the same conditions, a lower ACN content corresponds to a greater total crosslinking degree of the vulcanized rubber. This phenomenon is attributed to the higher ACN content resulting in fewer reaction sites for crosslinking and, consequently, a lower degree of crosslinking.

Moreover, increasing the filler content (from N5P5 to N6P5) enhances the entanglement between the rubber and filler, strengthening the rubber–filler network and increasing the number of physical crosslinking points. This, in turn, raises the torque difference of the HNBR composites. Conversely, increasing the plasticizer content (from P0 to P5) diminishes the interaction between the rubber macromolecular chains, leading to a significant reduction in the torque difference of the HNBR composites, which indicates a decrease in the total crosslinking degree.

Further investigations were conducted using a rubber processing analyzer to examine the effects of HNBR matrix type, fillers, and plasticizers on the rheological properties of HNBR compounds, as illustrated in Figure 3 and Figure 4.

The Payne effect describes the phenomenon where the dynamic storage modulus of the rubber matrix decreases rapidly with increasing strain when different fillers are incorporated. This effect is typically used to assess the network strength between fillers in the rubber matrix: a more pronounced Payne effect (indicated by a higher initial storage modulus) suggests stronger interactions between fillers, greater filler network strength, and more significant filler aggregation.

The results shown in Figure 3 indicate that adding five parts of plasticizer (from P0 to P5) significantly weakens the Payne effect in HNBR composites (Figure 3A: black line → red line; Figure 3B: black line → red line; Figure 3C: black line → red line and blue line → green line). This addition reduces filler aggregation, enhances filler dispersion within the HNBR matrix, and improves the processing performance of the rubber compound.

In contrast, increasing the filler content (from N5 to N6) significantly enhances the Payne effect in HNBR composites (Figure 3A: red line → blue line; Figure 3B: red line → blue line), increasing the strength of the filler–filler network and promoting greater filler aggregation.

Further research revealed that, under the same conditions, the type of HNBR matrix influences the Payne effect differently. After adding five parts of plasticizer, the decrease in storage modulus for various matrices is as follows: HNBR34 rubber composite (H3 matrix) exhibits ΔG3′ = 337 kPa, H4 matrix shows ΔG4′ = 422 kPa, H32 matrix has ΔG32′ = 224 kPa, and H42 matrix presents ΔG42′ = 409 kPa.

The impact of plasticizers on the loss factor of rubber primarily depends on the compatibility between the rubber matrix and the plasticizers. As shown in Figure 4, the addition of plasticizers (from P0 to P5) results in an increase in the loss factor of HNBR composites (Figure 4A: black line → red line; Figure 4B: black line → red line; Figure 4C: black line → red line and blue line → green line). This effect can be attributed to the small plasticizer molecules entering the spaces between the larger rubber molecular chains, which increases the distance between the chains and reduces their interactions. This leads to a significant decrease in the storage modulus, thereby increasing the loss factor.

Additionally, increasing the carbon black content (from N5 to N6) does not result in a significant change in the loss factor of HNBR composites (Figure 3A,B: red line → blue line). While carbon black can enhance the storage modulus of the rubber material, it also increases the loss modulus. Overall, the influence of carbon black content on the loss factor of HNBR composites is minimal.

### 3.2. Mechanical Properties of HNBR Composites

First, the effect of aging temperature on the stress–strain behavior of HNBR composites was investigated. The aging temperatures were set at 125 °C and 150 °C, and the stress–strain behavior was compared with that of unaged materials. The results are presented in Figure 5.

Figure 5 shows that after aging at high temperatures for 72 h, the stress–strain behavior of HNBR composites exhibits notable changes, particularly an increase in tensile stress, with HNBR43 composites showing the most significant increase. As the aging temperature rises from 125 °C to 150 °C, the fracture elongation, tensile strength, modulus at 100% (M100), and modulus at 200% (M200) of the HNBR composites vary as illustrated in Figure 6, Figure 7, Figure 8 and Figure 9.

For the non-aged HNBR composite, adding five parts of plasticizer (P0 → P5) significantly increases the elongation at break (purple column in Figure 6), while tensile strength experiences only a slight decrease (yellow column in Figure 7). Additionally, both the modulus at 100% (M100) (yellow column in Figure 8) and the modulus at 200% (M200) (purple column in Figure 9) show a notable downward trend. This occurs because the addition of plasticizer weakens the interaction between the rubber macromolecular chains, reducing the degree of crosslinking (indicated by the decrease in M100). Consequently, the relative slippage between macromolecular chains increases, leading to a decrease in tensile strength and an increase in elongation at break.

Conversely, increasing the amount of filler (N5 → N6) results in a significant decrease in the elongation at break of the HNBR composite material (purple column in Figure 6A,B), while tensile strength remains relatively unchanged (yellow column in Figure 7A,B). However, both M100 (yellow column in Figure 8A,B) and M200 (purple column in Figure 9A,B) increase significantly. This is attributed to carbon black acting as a rigid nano-reinforcing filler, forming a rubber–filler network that hinders the movement of macromolecular chains. Thus, increasing the filler content significantly raises tensile stress while greatly reducing elongation at break. The effects of plasticizer and filler on the performance of HNBR composites remain consistent across different aging temperatures (125 °C and 150 °C).

For the same type of HNBR composite, the influence of aging temperature on mechanical properties is more complex. Figure 6 and Figure 7 illustrate how aging temperature affects the fracture elongation and tensile strength of HNBR composites. After aging at 125 °C for 72 h, both fracture elongation and tensile strength exhibit slight increases. However, when the aging temperature rises to 150 °C, both parameters experience a decrease. The M100 value of HNBR composites shows a significant increase (Figure 8) with higher aging temperatures, while the increase in M200 value is relatively modest (Figure 9). This behavior is related to the amounts of plasticizer and filler in the formulation system of the HNBR composites.

### 3.3. Compression Set and Low-Temperature Performance of HNBR Composites

The rubber macromolecular chain undergoes certain physical or chemical changes when subjected to compression under varying conditions. Once the compressive force is removed, the rubber macromolecular chain experiences irreversible changes, leading to permanent set. To investigate the effects of temperature and medium on the compression set of HNBR composites, three distinct aging test conditions were selected: hot air aging at 125 °C for 24 h, low-temperature aging at −30 °C for 24 h, and hot oil aging at 125 °C for 168 h. The results are shown in Figure 10.

The results presented in Figure 10 indicate that temperature and medium conditions significantly influence the compression set of HNBR composites. HNBR composites exhibit excellent resistance to compression set in a 125 °C hot air environment but show poor resistance in low-temperature conditions (−30 °C cold air). This can be attributed to the fact that at low temperatures, which are below the glass transition temperature of HNBR, the rubber molecules become frozen. When the external compression force is released, the movement of these frozen molecules is restricted, leading to slow recovery and substantial permanent deformation at a macroscopic level.

In contrast to the performance in 125 °C hot air, HNBR composites experience greater compression set in a 125 °C hot transformer oil environment. This is primarily due to the diffusion of liquid oil molecules within the HNBR matrix, which is not significantly affected by the environmental temperature. At 125 °C, the temperature is much lower than the curing temperature of HNBR composites (175 °C), meaning it does not induce significant re-crosslinking of the rubber molecules, resulting in less permanent compression deformation. However, the movement rate of liquid oil molecules at 125 °C increases significantly, enhancing their diffusion rate in the HNBR matrix. The oil molecules infiltrate the spaces between the macromolecular chains, increasing the relative slippage between them and causing more substantial plastic deformation. Consequently, HNBR composites exhibit greater compression set in a 125 °C hot transformer oil environment.

Further analysis reveals that formula factors also impact the compression set of HNBR composites. HNBR34 composites, which have lower acrylonitrile content, demonstrate better resistance to compression set (Figure 10A), while higher acrylonitrile content in HNBR43 composites correlates with decreased resistance (Figure 10B). This trend relates to the mobility of HNBR macromolecular chains; higher acrylonitrile content increases polarity and intermolecular interaction forces, restricting chain movement.

The addition of five parts of plasticizer (P0 → P5) enhances the relative slippage between macromolecular chains, resulting in a significant increase in the compression set of HNBR composites. Conversely, increasing the filler content (N5 → N6) improves the resistance of HNBR composites to compression set, leading to reduced permanent deformation.

The low-temperature brittleness temperature of HNBR composites was determined using the multi-sample method, with results displayed in Table 3 and Table 4.

The brittleness temperature of rubber materials is influenced not only by the glass transition temperature of the rubber itself but also by the rubber’s strength and the composition of the formulation, particularly the type and amount of plasticizer used. Theoretically, a higher strength of rubber and a greater quantity of plasticizer correspond to a lower brittleness temperature.

As shown in Table 3 and Table 4, the low-temperature brittleness temperature of most HNBR composites is below −65 °C, with the exceptions being H4N5P0 and HNBR H32N5P0. Overall, HNBR composites demonstrate excellent low-temperature performance, which is closely linked to their high mechanical strength.

### 3.4. Oil Resistance Performance of HNBR Composites

HNBR exhibits exceptional oil resistance, aging resistance, ozone resistance, and chemical corrosion resistance, positioning it as one of the most remarkable specialty rubbers in terms of overall performance. It is extensively utilized in critical sectors such as automobile manufacturing, rail transportation, power grid systems, aerospace, and oil field development. Consequently, contact between HNBR products and liquid oil molecules is inevitable, leading to a certain degree of oil swelling, particularly at elevated temperatures. The extent of swelling in HNBR when exposed to oil is closely linked to the type of oil used. This article focuses on commonly used transformer oil as the research subject, investigating the changes in quality and hardness of HNBR composites in high-temperature transformer oil, as illustrated in Figure 11 and Figure 12.

From Figure 11, it is evident that after soaking in transformer oil at 125 °C for 168 h, the quality change rate of the HNBR43 composite, which has a higher acrylonitrile content, is the smallest (within 3%). In contrast, the HNBR34 composite with lower acrylonitrile content exhibits a larger degree of swelling (7–10%). The degree of swelling is closely related to the polarity of the HNBR matrix. A higher acrylonitrile content increases the polarity of HNBR, resulting in poorer compatibility with the non-polar transformer oil (due to the large polarity difference), which is reflected in the lower quality change rate. The polarity of HNBR and transformer oil can be characterized using three-dimensional solubility parameter values. By calculating the energy difference between the rubber and oil molecules, the swelling behavior of HNBR composites in transformer oil can be quantitatively analyzed and predicted [29,30,31,32].

Further analysis of the impact of plasticizers and fillers on the oil resistance of HNBR composites reveals that increasing the amount of plasticizer (P0 → P5) or filler (N5 → N6) results in a decrease in the rate of change in quality of the HNBR composites. For example, the impact of plasticizers on the rate of change in quality is illustrated as follows: 9.95% → 7.29% in Figure 11A, 2.74% → −0.34% in Figure 11B, and 12.23% → 9.54% and 8.31% → 5.51% in Figure 11C.

At high temperatures, some small molecules of plasticizers in the HNBR matrix diffuse into the transformer oil, leading to this decrease in the rate of change in quality of the HNBR composites, a phenomenon known as rubber extraction. Additionally, the incorporation of fillers obstructs the diffusion path of transformer oil molecules within the HNBR matrix, making the swelling of the rubber more difficult. Consequently, the rate of change in quality of HNBR composites also exhibits a decreasing trend.

In addition, the hardness changes of HNBR composites soaked in transformer oil at 125 °C for 168 h was studied, as shown in Figure 12. Hardness change is one of the most important performance evaluation indicators of rubber products during use. The results indicate that the hardness decrease of HNBR34 composites, which have a lower acrylonitrile content, is more significant. This is because the lower acrylonitrile content allows transformer oil to penetrate more easily into the rubber matrix, weakening the interaction between the rubber macromolecular chains and resulting in a more pronounced decrease in hardness (Figure 12A).

In contrast, HNBR43 composites with higher acrylonitrile content only experienced a slight change in hardness when exposed to high-temperature transformer oil, with the hardness either slightly decreasing or increasing (Figure 12B). Furthermore, the effects of plasticizers and fillers on the hardness of HNBR composites are opposite, reflecting the plasticizing effect of the plasticizers and the reinforcing effect of the fillers.

### 3.5. Ozone Resistance Characteristics of HNBR Composites

In the field of electric power applications within the power grid, HNBR sealing products are frequently exposed to the atmosphere and are susceptible to ozone attack, particularly in high-altitude areas where ozone concentrations are elevated. Therefore, it is essential to investigate the ozone resistance characteristics of HNBR composites. For this study, a high-concentration ozone environment was deliberately chosen for testing. The parameters were set as follows: a static elongation rate of 20%, an ozone concentration of 500 pphm, a temperature of 40 °C, and a testing duration of 16 h. The focus was on examining the surface cracking of HNBR composites, and the results are presented in Figure 13.

The results of the ozone resistance test shown in Figure 13 indicate that after aging in a high-concentration ozone environment, the surface of the HNBR composite did not exhibit any cracking. This suggests that the HNBR composite prepared in this experiment possesses excellent ozone resistance characteristics, making it suitable for use in high-altitude environments where ozone exposure is prevalent.

## 4. Conclusions

The structure of the rubber macromolecular chain significantly impacts the vulcanization characteristics of HNBR composites. HNBR34, which has a lower acrylonitrile (ACN) content, demonstrates higher torque and a greater degree of crosslinking. Increasing the filler content leads to an increase in both the maximum and minimum torque of the HNBR composites, with a stronger rubber–filler network structure enhancing the physical crosslinking degree of HNBR.

Plasticizers play a crucial role by reducing the interaction between rubber macromolecular chains, having a pronounced effect on vulcanization torque. For instance, the addition of five parts of plasticizer can significantly decrease the maximum torque and the overall crosslinking degree of HNBR composites. Notably, the effects of plasticizers and fillers on the Payne effect of HNBR composites are opposite: plasticizers weaken the Payne effect, thereby improving the processing performance, while an increase in filler content significantly enhances the Payne effect due to a strengthened filler–filler network.

The addition of plasticizers also leads to a notable increase in the elongation at break of HNBR composites, although it reduces both the M100 and M200 values, with minimal impact on tensile strength. Conversely, increasing the filler content results in a significant decrease in elongation at break, while M100 and M200 increase, with tensile strength remaining largely unchanged.

The influence of aging temperature on the mechanical properties of HNBR composites is complex and is closely tied to the amount of plasticizers and fillers in the formulation. The compression set of HNBR composites is markedly influenced by temperature and medium conditions, showing superior resistance to compression set in a 125 °C hot air environment compared to a low-temperature (−30 °C) environment. Additionally, the compression set in a 125 °C hot transformer oil environment is also relatively high.

Regarding formulation factors, HNBR composites with lower acrylonitrile content exhibit less compression set. Plasticizers significantly increase compression set, while increasing filler content can reduce it. The rate of change in quality of HNBR composites in transformer oil inversely correlates with the increase in acrylonitrile content. Furthermore, increasing the plasticizer or filler content leads to a decrease in the rate of change in quality.

Lastly, high-concentration ozone aging crack experiments indicate that HNBR composites possess significant application value and performance advantages in high-altitude environments.

## Figures and Tables

**Figure 1 polymers-16-03294-f001:**
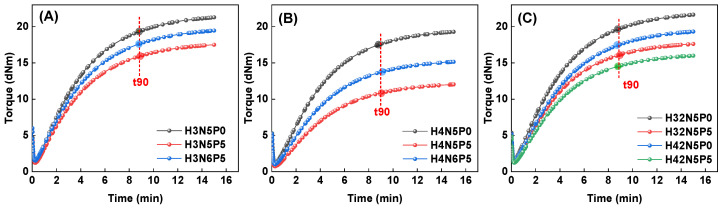
Curing curve of HNBR composites: (**A**) HNBR34 composites, (**B**) HNBR43 composites, (**C**) HNBR34/HNBR21 and HNBR43/HNBR21 composites.

**Figure 2 polymers-16-03294-f002:**
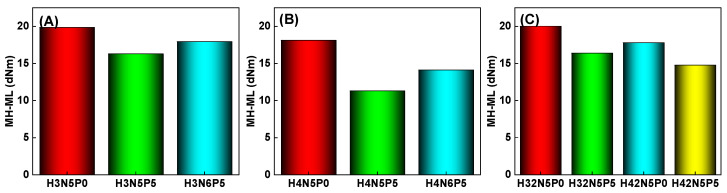
Crosslinking degree (MH-ML) of HNBR composites: (**A**) HNBR34 composite, (**B**) HNBR43 composite, (**C**) HNBR34/HNBR21 and HNBR43/HNBR21 composites.

**Figure 3 polymers-16-03294-f003:**
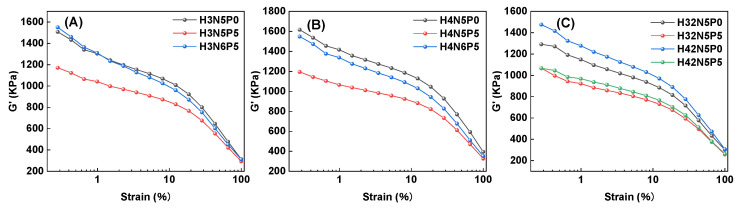
The relationship between the storage modulus (G’) and strain of HNBR compounds: (**A**) HNBR34 compound, (**B**) HNBR43 compound, (**C**) HNBR34/HNBR21 and HNBR43/HNBR21 compounds.

**Figure 4 polymers-16-03294-f004:**
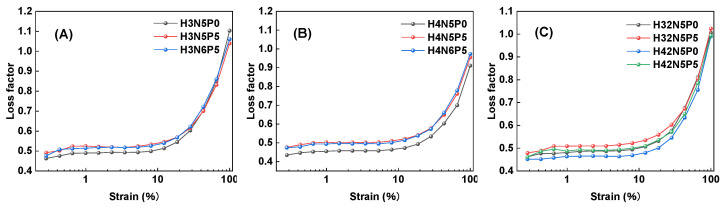
Relationship between loss factor and strain of HNBR compounds: (**A**) HNBR34 compound, (**B**) HNBR43 compound, (**C**) HNBR34/HNBR21 and HNBR43/HNBR21 compounds.

**Figure 5 polymers-16-03294-f005:**
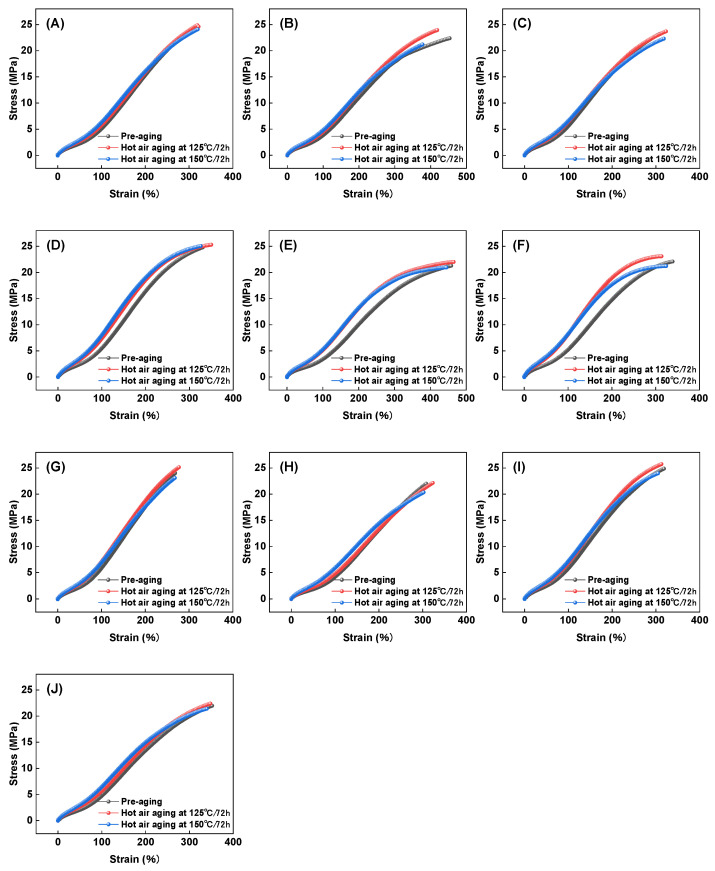
Stress–strain curves of HNBR composites: (**A**) H3N5P0, (**B**) H3N5P5, (**C**) H3N6P5, (**D**) H4N5P0, (**E**) H4N5P5, (**F**) H4N6P5, (**G**) H32N5P0, (**H**) H32N5P5, (**I**) H42N5P0, (**J**) H42N5P5.

**Figure 6 polymers-16-03294-f006:**
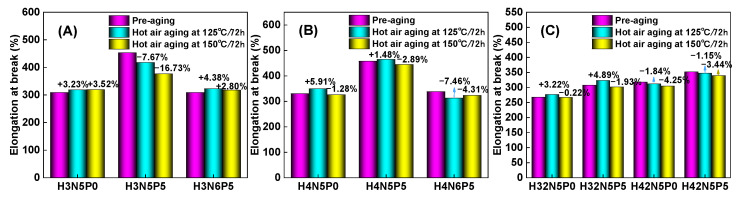
Effect of aging temperature on the fracture elongation of HNBR composites: (**A**) HNBR34 composite, (**B**) HNBR43 composite, (**C**) HNBR34/HNBR21 and HNBR43/HNBR21 composites.

**Figure 7 polymers-16-03294-f007:**
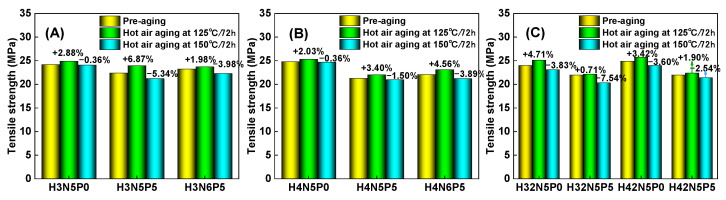
Effect of aging temperature on the tensile strength of HNBR composites: (**A**) HNBR34 composite, (**B**) HNBR43 composite, (**C**) HNBR34/HNBR21 and HNBR43/HNBR21 composites.

**Figure 8 polymers-16-03294-f008:**
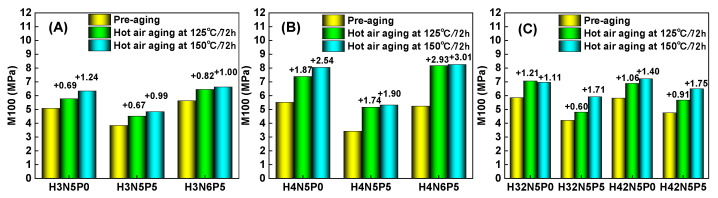
Effect of aging temperature on the 100% tensile stress of HNBR composites: (**A**) HNBR34 composite, (**B**) HNBR43 composite, (**C**) HNBR34/HNBR21 and HNBR43/HNBR21 composites.

**Figure 9 polymers-16-03294-f009:**
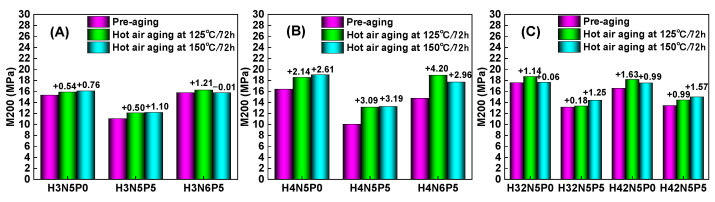
Effect of aging temperature on the 200% tensile stress of HNBR composites: (**A**) HNBR34 composite, (**B**) HNBR43 composite, (**C**) HNBR34/HNBR21 and HNBR43/HNBR21 composites.

**Figure 10 polymers-16-03294-f010:**
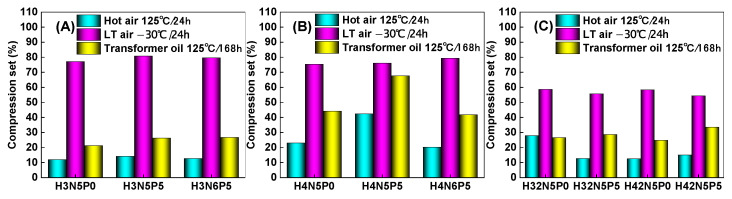
Effect of aging conditions on the compression set of HNBR composites: (**A**) HNBR34 composite, (**B**) HNBR43 composite, (**C**) HNBR34/HNBR21 and HNBR43/HNBR21 composites.

**Figure 11 polymers-16-03294-f011:**
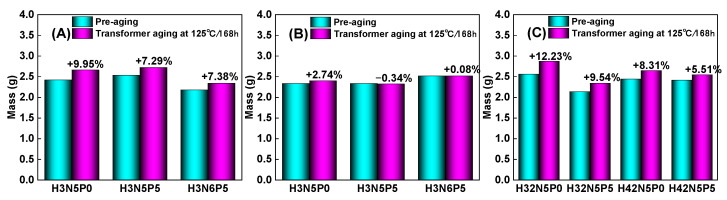
Mass change of HNBR composites in transformer oil (125 °C/168 h): (**A**) HNBR34 composite, (**B**) HNBR43 composite, (**C**) HNBR34/HNBR21 and HNBR43/HNBR21 composites.

**Figure 12 polymers-16-03294-f012:**
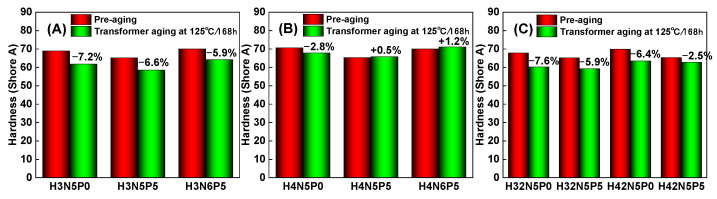
Hardness change of HNBR composites in transformer oil (125 °C/168 h): (**A**) HNBR34 composite, (**B**) HNBR43 composite, (**C**) HNBR34/HNBR21 and HNBR43/HNBR21 composites.

**Figure 13 polymers-16-03294-f013:**
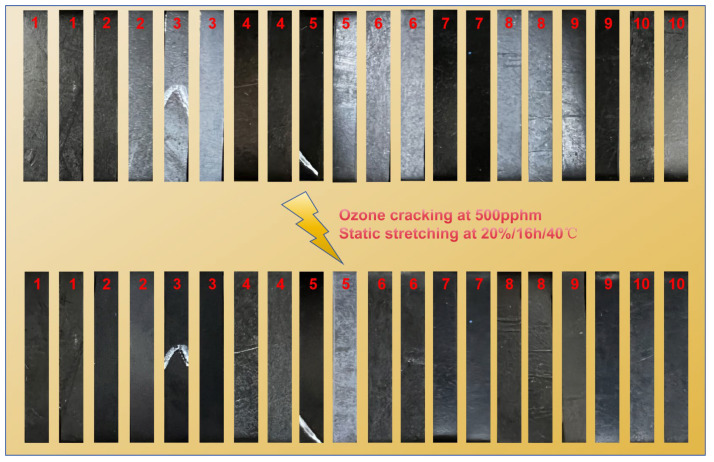
Ozone cracking resistance of HNBR composites: (1) H3N5P0, (2) H3N5P5, (3) H3N6P5, (4) H4N5P0, (5) H4N5P5, (6) H4N6P5, (7) H32N5P0, (8) H32N5P5, (9) H42N5P0, (10) H42N5P5.

**Table 1 polymers-16-03294-t001:** Formulation of HNBR monomer composite material (mass fraction).

Ingredients, phr	H3N5P0	H3N5P5	H3N6P5	H4N5P0	H4N5P5	H4N6P5
HNBR34	100	100	100	0	0	0
HNBR43	0	0	0	100	100	100
Carbon black	50	50	60	50	50	60
Plasticizer	0	5	5	0	5	5

Other ingredients in the formula: 6 parts of peroxide crosslinking agent, 2 parts of crosslinking agent, 5 parts of zinc oxide, 1 part of stearic acid, 1 part of antioxidant. Using the codes H3N5P0 and H4N6P5 as examples, their meanings are as follows: “H3” denotes the HNBR34 rubber matrix, and “H4” refers to the HNBR43 matrix. “N5” indicates a carbon black content of 50 parts, while “N6” corresponds to 60 parts. “P0” represents a plasticizer content of 0 parts, and “P5” signifies a plasticizer content of 5 parts.

**Table 2 polymers-16-03294-t002:** Formulation of HNBR and matrix composite materials (mass fraction).

Ingredients, phr	H32N5P0	H32N5P5	H42N5P0	H42N5P5
HNBR34	60	60	0	0
HNBR43	0	0	60	60
HNBR21	40	40	40	40
Carbon black	50	50	50	50
Plasticizer	0	5	0	5

Other ingredients in the formula: 6 parts of peroxide crosslinking agent, 2 parts of crosslinking agent, 5 parts of zinc oxide, 1 part of stearic acid, 1 part of antioxidant. Using the codes H32N5P0 and H42N5P5 as examples, their meanings are as follows: “H32” denotes a composite matrix of HNBR34 and HNBR21 rubber, while “H42” represents a composite matrix of HNBR43 and HNBR21. “N5” indicates a carbon black content of 50 parts, and “N6” corresponds to 60 parts. “P0” represents a plasticizer content of 0 parts, and “P5” indicates a plasticizer content of 5 parts.

**Table 3 polymers-16-03294-t003:** Brittle temperature of HNBR composites (1).

Formula	H3N5P0	H3N5P5	H3N6P5	H4N5P0	H4N5P5	H4N6P5
Brittleness temperature	≤−65 °C	≤−65 °C	≤−65 °C	−65 °C ≤ −60 °C	≤−65 °C	≤−65 °C

**Table 4 polymers-16-03294-t004:** Brittle temperature of HNBR composites (2).

Ingredients, phr	H32N5P0	H32N5P5	H42N5P0	H42N5P5
Brittleness temperature	−65 °C ≤ −60 °C	≤−65 °C	≤−65 °C	≤−65 °C

## Data Availability

The data presented in this study are available in the article.

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
