# Peer review of "Investigation of Mechanical Properties and Oil Resistance of Hydrogenated-Butadiene-Acrylonitrile-Rubber-Based Composites Across Various Temperatures"

_polymers, 2024, doi:10.3390/polym16233294_

Round 1

Reviewer 1 Report

Comments and Suggestions for Authors

This is the effect of simultaneous addition of CB and oil on the mechanical properties for HNBR-based compounds. Since HNBR is focused recently, it is an important work for industrial applications. However, the followings are needed to discuss further to be published in the journal.

1. The details of raw materials should be mentioned more: (1) What are DCP and TAIC? (2) What is the plasticizer? (3) Molecular weight of the rubbers, (4) degree of saturation, or diene content, (5) what is the antioxidant? And so on.

2. Figure 3: (1) how can they evaluate the G’, i.e., linear values, in the non-linear region? (2) All data showed the strain dependence, meaning no linear region. Therefore, the authors cannot discuss the strength of the Payne effect. Show the linear values. (3) The unit should be kPa, not KPa. 

3. The same question on Figure 2.

4. Aging effect should be discussed based on the swelling ratio.

Reviewer 2 Report

Comments and Suggestions for Authors

I have the following comments regarding this article:

-          The affiliations are not complete.

-          Please characterize in the text what is chemical structure of the plasticizer TP-95, co-crosslinking agent TAIC, and  antioxidant 445. It is crucial to know this in order to perform valid results interpretation and discussion.

-          What is the role of zinc oxide and stearic acid in the formulations? Normally they are activators for sulfur vulcanization, however, they don’t play any role in the peroxide vulcanization.

-          Please, add more detailed description of the mixer, mill, and press used for the samples preparation.

-          Please increase the font in the figures – in most cases the font is too low to be readable.

-          The authors explain lowering crosslinking degree with the lowering ACN concentrations by high elecronegativity of the ACN group (lines 166-169). However, in my opinion, this is simply related with higher content of hydrogenated butadiene mers, which predominantly undergo peroxide cross-linking. The lower the CAN content the higher the hydrogenated butadiene content resulting in higher crosslinking degree.

-          Were the rheological properties (strain sweep) of the compounds measured on vulcanized of unvulcanized samples? Please describe in the text.
